# K-Space Approach in Optical Coherence Tomography: Rigorous Digital Transformation of Arbitrary-Shape Beams, Aberration Elimination and Super-Refocusing beyond Conventional Phase Correction Procedures

**DOI:** 10.3390/s24092931

**Published:** 2024-05-05

**Authors:** Alexander L. Matveyev, Lev A. Matveev, Grigory V. Gelikonov, Vladimir Y. Zaitsev

**Affiliations:** A.V. Gaponov-Grekhov Institute of Applied Physics RAS, Nizhny Novgorod 603950, Russia; matveyev@ipfran.ru (A.L.M.); lev@ipfran.ru (L.A.M.); grgel@ipfran.ru (G.V.G.)

**Keywords:** optical coherence tomography, K-space representation, angular spectrum, digital refocusing, digital image transformation, non-paraxial approximation, non-Gaussian beams, aberration correction

## Abstract

For the most popular method of scan formation in Optical Coherence Tomography (OCT) based on plane-parallel scanning of the illuminating beam, we present a compact but rigorous K-space description in which the spectral representation is used to describe both the axial and lateral structure of the illuminating/received OCT signals. Along with the majority of descriptions of OCT-image formation, the discussed approach relies on the basic principle of OCT operation, in which ballistic backscattering of the illuminating light is assumed. This single-scattering assumption is the main limitation, whereas in other aspects, the presented approach is rather general. In particular, it is applicable to arbitrary beam shapes without the need for paraxial approximation or the assumption of Gaussian beams. The main result of this study is the use of the proposed K-space description to analytically derive a filtering function that allows one to digitally transform the initial 3D set of complex-valued OCT data into a desired (target) dataset of a rather general form. An essential feature of the proposed filtering procedures is the utilization of both phase and amplitude transformations, unlike conventionally discussed phase-only transformations. To illustrate the efficiency and generality of the proposed filtering function, the latter is applied to the mutual transformation of non-Gaussian beams and to the digital elimination of arbitrary aberrations at the illuminating/receiving aperture. As another example, in addition to the conventionally discussed digital refocusing enabling depth-independent lateral resolution the same as in the physical focus, we use the derived filtering function to perform digital “super-refocusing.” The latter does not yet overcome the diffraction limit but readily enables lateral resolution several times better than in the initial physical focus.

## 1. Introduction

Various methods are used to describe and numerically simulate image formation in optical coherence tomography (OCT), among which much attention has been paid to such a powerful approach as Monte-Carlo (MC), the basics of which were recently discussed in overview [1] and its direct application to OCT can be found, e.g., in [2,3]. The strength of MC-methods is related to the very MC principle, in which the image is formed due to a huge number of scattering events of individual photons. This principle implies a rather high computational demand, but at the same time makes the MC approach rather general. In particular, the MC approach allows for the accounting of multiple scattering, which is difficult to introduce in other approaches.

It can be recalled that the principle of OCT-image formation [4,5,6] is based on ballistic single backscattering of the illuminating optical beam (i.e., backscattering described in the first-order Born approximation), so that, among various reasons for OCT-image degradation, multiple scattering is only one such factor, consideration of which for many applications is not of primary importance. In view of this, significant attention was paid to the development of models of OCT-scan formation based on the wave representation and single backscattering of the illuminating beams. In such wave-based approaches, the illuminating field of OCT systems can be described using somewhat different representations. In some cases, important features of OCT-image formation could be understood using simple plane-wave approximations in various forms [7,8]. For example, image formation in spectral-domain OCT can be described by representing the illuminating field as a set of co-propagating plane waves with different axial wavenumbers. This approximation, even without special attention to the lateral beam structure, can be sufficient to describe the axial form of the point-spread function (PSF). In some variants, along with the description of the PSF form in the axial direction by considering a set of spectral components with different axial wavenumbers, the lateral form of the OCT beam was also taken into account using the spatial-domain description. Often, the lateral field distribution was described using Gaussian beams [9,10,11,12]. Such approaches opened the possibility of fairly accurately describing the lateral form of PSF, including the influence of the illuminating beam focusing. However, such Gaussian-beam-based models do not allow one to describe other beam forms (e.g., Bessel beams), which also represent interest for utilization in OCT [13,14,15].

Alternatively, for describing the lateral distribution of the illuminating and scattered fields, other models based on angular spectra (i.e., representation of the field as a set of plane waves with various directions) have also been developed with one or another degree of rigor (see, e.g., [16,17,18,19,20]). Such approaches combine the spectral representation of both the axial and lateral structures of illuminating beams and, therefore, can be termed a full K-space description. Recently, a comprehensive overview of various K-space-based approaches developed in various techniques of optical visualization (not only OCT in the conventional sense) was published [21]. It was pointed out in [21] that the K-space description is a rather general approach, allowing one to analyze very diverse factors in a unified manner, including various beam shapes, effects of dispersion, and aberrations of the optical-wave front.

In application to conventional point-scanning OCT systems, adequate post-processing of the acquired 3D sets of complex-valued OCT signals opens flexible possibilities for various transformations of the initial OCT data. In particular, due to such processing, it becomes possible to introduce corrections for PSF distortions caused by aberrations and/or dispersion effects during wave propagation [21]. It also becomes possible to perform digital refocusing by reassembling the spectral components of the illuminating field with corrected phases, such that the depth-independent focusing can be formed over the entire 3D dataset (see examples of such digital transformations in [19,21]). Similarly, the problem of digital corrections of aberrations was discussed (e.g., [21,22]), so that the quality of images obtained by an OCT system with an inexpensive optical system and appreciable aberrations can be made comparable with the quality of images obtained with a much more expensive aberration-less optics.

Here, using the terminology of paper [21], we consider a modified version of the K-space description of OCT-image formation in the development of recent paper [19]. The main limitation of this approach is the use of single scattering, which is a generally accepted approximation [6,20,21] corresponding to the very principle of OCT operation. We analyze the most common point-scanning scheme of OCT. In this scheme, the illuminating and then backscattered optical signal passes forth-and-back through the tissue and the same illuminating/receiving aperture. In particular, in the presence of aberrations at this aperture, the optical field passes twice through the source of aberrations, so that the aberration-related distortions in the illuminating and scattered fields experience nonlinear mixing at the reception. The forth-and-back propagation of the optical wave is considered in the K-space (angular-spectrum) representation rigorously without such often-used simplifying limitations as the paraxial approximation or utilization of Gaussian beams. Even if the derived integral expressions without such approximations cannot be evaluated analytically, they can readily be estimated numerically for arbitrary shapes of illuminating beams, not necessarily possessing circular symmetry, e.g., in the presence of anisotropic aberrations, as demonstrated below.

The main goal of the present study is the use of the K-space approach, basically similar to [19], to derive a general form of a filtering function that is intended to transform the initial 3D set of complex-valued OCT data into a “desired”(or “target”) form. The latter can be understood in a rather general sense. An essential feature of the proposed filtering procedures is the utilization of both phase and amplitude transformations, unlike often discussed procedures based on only phase transformations. After presenting the basic equations describing the initial OCT-signal formation and deriving the general form of the filtering function, we illustrate the generality and efficiency of the proposed approach by considering several instructive examples. In particular, we demonstrate a non-Gaussian example of the transformation of a 3D set of OCT data (initially corresponding to a beam with a Lorentzian profile to the 3D image for a Bessel beam). Examples will be given to demonstrate that the influence of arbitrary (including anisotropic) aberrations at the illuminating/receiving aperture can be digitally compensated (despite the nonlinear mixing of aberrations acquired during forward and backward propagation). Next, in addition to conventionally discussed digital refocusing intended to obtain the same lateral resolution as at the depth of physical focus, we consider an example of digital filtering enabling “super refocusing”. This super-refocusing does not yet overcome the diffraction limit imposed by the illuminating-light wavelength, but allows one to overcome the lateral resolution limited by the initial radius of the focal waist of the illuminating beam, although the increased resolution in such super-refocused images is obtained at the expense of some reduction in the signal-to-noise ratio (SNR).

## 2. Rigorous K-Space Formulation of OCT-Image Formation in the 1st-Order Born Approximation of the Scattered Signal

The spectral approach in various forms has been widely used for describing OCT-image formation (e.g., [16,17,18,21,23]). In what follows, we will use the K-space formulation presented in [19] as a basis for the further analysis. In the framework of ballistic first-order scattering (which corresponds to the 1st-order Born approximation), this description is rather rigorous and does not need other popular approximations, such as the widely used paraxial approximation or mandatory utilization of Gaussian beams.

Each spectral component of the illuminating beam with wavenumber k is characterized by a complex-amplitude distribution $U_{L}(x-x_{0},y-y_{0};k)$ at the output lens aperture with the axis passing through lateral coordinates $\left(x_{0},y_{0}\right)$. The source is characterized by the discrete spectrum $S(k=k_{n})$ over the wavenumber. The lens is orthogonal to *z*-axis, corresponding to the axial direction, the lens plane has the axial coordinate $z=z_{d}$ as shown in Figure 1. Coordinate $z_{d}$ is the distance between the aperture and the tissue boundary (and often this gap is filled with an immersion layer). Scanning of the beam position with lateral coordinates $\left(x_{0},y_{0}\right)$ is used for obtaining the 3D volume of OCT data. We also use a widely utilized approximation of discrete scatterers [24] that are characterized by coordinates $\left(x_{s},y_{s},z_{s}\right)$. Such localized scatterers are convenient for representing the initial and transformed images, but the discussed spectral transformations of optical fields are rigorous and independent of the scatterer density. Like in real-spectral-domain OCT, we represent the illuminating beam as a set of spectral components with wavenumbers $k=k_{n}$.

For a given wavenumber k, the distribution $U_{L}(x-x_{0},y-y_{0};k)$ of the complex-valued field over the input/output lens with the axis passing through lateral coordinates $\left(x_{0},y_{0}\right)$ is characterized by the angular spectrum ${\hat{U}}_{L}(x_{0},y_{0};k_{x},k_{y},k)$ defined by the following Fourier-transform over lateral coordinates (x,y):(1)$${\hat{U}}_{L}(x_{0},y_{0};k_{x},k_{y},k)=\iint U_{L}(x-x_{0},y-y_{0};k)e^{-i(k_{x}x+k_{y}y)}dxdy=g(k_{x},k_{y},k)e^{-ix_{0}k_{x}-iy_{0}k_{y}}$$
where $k_{x},k_{y}$ are the lateral components of the total wavenumber k; $g(k_{x},k_{y},k)$ is the angular spectrum of the illuminating-beam distribution $U_{L}(x-x_{0},y-y_{0};k)$ over lateral coordinates (x,y) for the axis position $\left(x_{0}=0,y_{0}=0\right)$. The representation of the illuminating beam via its angular spectrum $g(k_{x},k_{y},k)$ has the advantage that it allows one to describe an arbitrary beam profile as a set of plane waves with the wave vector components $\left(k_{x},k_{y},k_{z}=\sqrt{k^{2}-k_{x}^{2}-k_{y}^{2}}\right)$.

During propagation between the input-lens plane $z=z_{d}$ and plane $z=z_{s}$ corresponding to the depth of *s*-th scatterer each of the plane-wave components acquires the additional phase $(z_{s}-z_{d})\sqrt{k^{2}-k_{x}^{2}-k_{y}^{2}}$, which means that the complex-valued amplitude of each propagating components is multiplied by the following propagator function $h_{s}(k_{x},k_{y},k)$:(2)$$h_{s}(k_{x},k_{y},k)=e^{i(z_{s}-z_{d})\sqrt{k^{2}-k_{x}^{2}-k_{y}^{2}}}$$

Here, subscript “*s*” means that $h_{s}$ depends on the depth $z_{s}$ of *s*-th scatterer and it is assumed that exp(ikz) corresponds to propagation along *z*-axis (i.e., in-depth propagation). Therefore, according to Equations (1) and (2), the angular spectrum ${\hat{U}}_{s}(x_{0},y_{0},k_{x},k_{y},k)$ of the field propagated from the illuminating aperture with the lateral coordinates $\left(x_{0},y_{0}\right)$ to the location of the *s*-th scatterer with coordinates $\left(x_{s},y_{s},z_{s}\right)$ is given the following product
(3)$${\hat{U}}_{s}(x_{0},y_{0},k_{x},k_{y},k)=h_{s}(k_{x},k_{y},k)g(k_{x},k_{y},k)e^{-ik_{x}x_{0}-ik_{y}y_{0}}$$

Notice that spectrum ${\hat{U}}_{s}(x_{0},y_{0},k_{x},k_{y},k)$ in Equation (3) depends only on the axial coordinate of the scatterer via function $h_{s}(k_{x},k_{y},k)$. In contrast, lateral coordinates $\left(x_{s},y_{s}\right)$ of *s*-th scatterer do not enter the incident-wave spectrum (3). Its inverse Fourier transform over spectral coordinates $\left(k_{x},k_{y}\right)$ gives the complex amplitude $U_{s}(x_{0},y_{0};k)$ of the incident beam with lateral coordinates $\left(x_{0},y_{0}\right)$ in the location $\left(x_{s},y_{s},z_{s}\right)$ of the scatterer:(4)$$\begin{array}{l} U_{s}(x_{0},y_{0};k)=\iint_{k_{x}^{2}+k_{y}^{2}\leq k^{2}}{\hat{U}}_{s}(x_{0},y_{0};k_{x},k_{y},k)e^{i(k_{x}x_{s}+k_{y}y_{s})}dk_{x}dk_{y}=\\ =\iint_{k_{x}^{2}+k_{y}^{2}\leq k^{2}}g(k_{x},k_{y},k)h_{s}(k_{x},k_{y},k)e^{i[k_{x}(x_{s}-x_{0})+k_{y}(y_{s}-y_{0})]}dk_{x}dk_{y} \end{array}$$

We emphasize that subscript “s” in $U_{s}(x_{0},y_{0};k)$ means that the incident-wave amplitude depends not only on the position of the beam axis $\left(x_{0},y_{0}\right)$, but also on coordinates $\left(x_{s},y_{s},z_{s}\right)$ of the *s*-th scatterer.

Propagation of the backscattered signal can be conveniently represented, as shown in Figure 2, in which the back-propagation path is mirrored, as if the back-propagated signal effectively propagates in the same direction along the *Z*-axis. In this representation, the symmetry between forward and back propagation is especially clear. The distance of the two paths is exactly the same, and the axial back-propagation from the *s*-th scatterer towards the lens again corresponds to multiplication by the propagator $h_{s}(k_{x},k_{y},k)$ given by Equation (2). Next, for each position $\left(x_{0},y_{0}\right)$ of the lens axis, when passing through the lens, the amplitudes of received spectral components are multiplied by the factor $g(k_{x},k_{y},k)e^{-ix_{0}k_{x}-iy_{0}k_{y}}$ given by Equation (1). Thus, the spectrum transformation during the back propagation is described by the same total spectral factor $h_{s}(k_{x},k_{y},k)g(k_{x},k_{y},k)e^{-ix_{0}k_{x}-iy_{0}k_{y}}$ as in Equation (3) for the forward propagation. Consequently, the transition from the spectral representation to the received signal amplitude is again given by the Fourier transform, similar to Equation (4).

It should also be taken into account that in the plane $z=z_{s}$ passing through *s*-th scatterer the spectral components of the scattered signal with the wavenumber k can be represented as $K_{s}\cdot U_{s}(x_{0},y_{0};k)$, where the scattering coefficient $K_{s}$ characterizes the proportionality between the scattered-wave amplitude and the amplitude $U_{s}(x_{0},y_{0};k)$ of the wave incident onto the scatterer located at $\left(x_{s},y_{s},z_{s}\right)$. Notice that coefficient $K_{s}$ may be different for different scatterers. Taking this remark into account the amplitude of the received signal $B_{s}(x_{0},y_{0};k)$ from *s*-th scatterer can be written similarly to Equation (4):(5)$$B_{s}(x_{0},y_{0};k)=K_{s}U_{s}(x_{0},y_{0};k)\iint_{k_{x}^{2}+k_{y}^{2}\leq k^{2}}g(k_{x},k_{y},k)h_{s}(k_{x},k_{y},k)e^{i[k_{x}(x_{s}-x_{0})+k_{y}(y_{s}-y_{0})]}dk_{x}dk_{y}$$
so that recalling Equation (4) for $U_{s}(x_{0},y_{0};k)$, one can rewrite Equation (5) in a simple and compact form
(6)$$B_{s}(x_{0},y_{0};k)=K_{s}\cdot U_{s}^{2}(x_{0},y_{0};k)$$

Here, we assume that for widely used OCT systems with a fairly narrow bandwidth of the illuminating source of the order of several percent and for nearly point-like scatterers with broad scattering diagrams, the dependence of the coefficient $K_{s}$ on the wavenumber k and wavenumber projections $\left(k_{x},k_{y}\right)$ can the neglected. Consequently, $K_{s}$ can be considered approximately constant. However, if necessary, the above-mentioned effects can be incorporated into the dependence $K_{s}=K_{s}(k_{x},k_{y},k)$ and this coefficient should enter the Fourier integral in Equation (5). It should also be emphasized that the symmetry of the received and incident at *s*-th scatterer signals in Equations (5) and (6) holds for beams of arbitrary form, not only Gaussian ones.

To find the total form of an A-scan corresponding to the position of the illuminating-beam axis $\left(x_{0},y_{0}\right)$ the contribution of all scatterers should be summed by performing summation over index s of scatterers, and the (inverse) Fourier transform should be performed to combine the contributions of all received spectral components with the wavenumbers $k=k_{n}$:(7)$$A\left(x_{0},y_{0},z_{q}\right)=DFT_{k_{n}\to z_{q}}\left\{\sum_{s}S(k_{n})B_{s}\left(x_{0},y_{0};k_{n}\right)\right\}$$

Here, the abbreviation DFT (Digital Fourier Transform) emphasizes that this Fourier transform is made numerically, for example, using efficient *FFT* algorithms realized in many numerical packages; the additionally introduced function $S(k_{n})$ describes the amplitudes of spectral components with wavenumbers $k=k_{n}$ of the illuminating-source spectrum.

Equations (1)–(7) constitute a convenient, computationally highly efficient framework for simulating various types of OCT scans for exactly controllable and flexibly variable conditions. Depending on the particular problem, the simulated visualized regions with realistic millimeter-scale sizes may contain a few scatterers for demonstration. Also, configurations with $>10^{5}$ scatterers separated by several micrometers to imitate the spatial density of scatters typical of real biological tissues can be readily simulated, as was recently demonstrated in [19]. In particular, simulations of regions with either regularly or stochastically moving scatters can efficiently be made, for which alternative methods (such as Monte-Carlo-based ones) are not well suitable.

Concerning the further discussed procedures for digital transformation of 3D sets of OCT data composed of A-scans given by Equation (7), it should be noted that for a given position $\left(x_{0},y_{0}\right)$ of the illuminating-beam axis, we obtain only one 1D A-scan $A\left(x_{0},y_{0},z_{q}\right)$ given by Equation (7). The latter contains complex amplitudes $B_{s}(x_{0},y_{0};k_{n})$ of the received wave components with wavenumbers $k_{n}$. Amplitudes $A\left(x_{0},y_{0},z_{q}\right)$ and $B_{s}(x_{0},y_{0};k_{n})$ received for a given position $\left(x_{0},y_{0}\right)$ of the beam axis do not yet give information about the distribution of amplitude and phase of the OCT signal in the lateral directions, which is indispensable if we want to digitally transform the OCT image (e.g., for performing digital refocusing). In the K-space description, such transformations correspond to the reassembling of angular spectral components of the received signals. Therefore, to find the set of spectral components ${\hat{B}}_{s}(k_{x},k_{y},k_{n})=DFT_{x_{0},y_{0}\to k_{x},k_{y}}[B_{s}(x_{0},y_{0};k_{n})]$ one needs to perform scanning of the illuminating beam in *x*- and y-directions to obtain a sufficiently large and densely spaced set of acquired complex-valued A-scans $B_{s}(x_{0},y_{0};k_{n})$ for various $\left(x_{0},y_{0}\right)$. These coordinates should be equidistantly spaced to efficiently perform the Fourier transform from the spatial domain to the K-space. Next, to enable the desired lateral resolution, the initial set of spatial data should be sufficiently dense to satisfy the Nyquist-Kotelnikov criterion. In particular, to enable OCT-image refocusing for a focused beam, the distance between adjacent positions of A-scans in both lateral directions should be smaller than the focal radius of the beam. The corresponding examples will be given in the following sections.

## 3. A Remark about Independence of Manipulations with the Angular Spectrum of OCT Signals on the Lateral Positions of Scatterers

In what follows, we perform various transformations (such as refocusing, etc.) with the acquired 3D sets of OCT data. For practical usage, a key point is that, for such transformation of OCT images as refocusing, the procedures certainly should account for the specific depth but should not require *a priori* knowledge of the lateral positions of scatterers. Mathematically, this means that when performing the corresponding manipulations in K-space with OCT data, the lateral positions of scatterers can be factorized independently of the axial positions of these scatterers.

To demonstrate this, we consider Equation (7) describing the complex-valued A-scans in the acquired 3D-pack of OCT data. To manipulate the complex-valued quantities $A\left(x_{0},y_{0},z_{q}\right)$ in K-space, they should be transformed to the spectral form by applying the Fourier transform $DFT_{x_{0},y_{0}\to k_{x},k_{y}}$ to Equation (7). Since the Fourier transform $DFT_{x_{0},y_{0}\to k_{x},k_{y}}$, as well as the summations and the inverse Fourier transform $DFT_{k_{n}\to z_{q}}$ in Equation (7) are linear operations, their order can be arbitrary. Therefore, in Equation (7), we can apply $DFT_{x_{0},y_{0}\to k_{x},k_{y}}$ to amplitudes $B_{s}(x_{0},y_{0};k)$ of the received signal from *s*-th scatterer: ${\hat{B}}_{s}(k_{x},k_{y},k)=DFT_{x_{0},y_{0}\to k_{x},k_{y}}\{B_{s}(x_{0},y_{0};k)\}$. The structure of $B_{s}(x_{0},y_{0},k)$ is given by Equation (6). Bearing in mind that the Fourier transform of a product corresponds to the convolution of the Fourier-transform of the multipliers, one can write:(8)$$\begin{array}{l} {\hat{B}}_{s}(k_{x},k_{y},k)=DFT_{x_{0},y_{0}\to k_{x},k_{y}}\{B_{s}(x_{0},y_{0};k)\}=K_{s}\cdot DFT_{x_{0},y_{0}\to k_{x},k_{y}}\{U_{s}^{2}(x_{0},y_{0};k)\}=\\ =K_{s}DFT_{x_{0},y_{0}\to k_{x},k_{y}}\{U_{s}(x_{0},y_{0};k)\}⊗DFT_{x_{0},y_{0}\to k_{x},k_{y}}\{U_{s}(x_{0},y_{0};k)\} \end{array}$$

Recalling that $U_{s}(x_{0},y_{0};k)$ is given by Equation (4), one can notice that in Equation (8), the convolution can be explicitly represented in the following form
(9)$${\hat{B}}_{s}(k_{x},k_{y},k)=DFT_{x_{0},y_{0}\to k_{x},k_{y}}\{B_{s}(x_{0},y_{0};k)\}=K_{s}e^{-i[k_{x}x_{s}+k_{y}y_{s}]}I(z_{s};k,k_{x},k_{y})$$
where we denote
(10)$$I(z_{s};k,k_{x},k_{y})=\iint g(\xi ,\eta ,k)h_{s}(\xi ,\eta ,k)g(-k_{x}-\xi ,-k_{y}-\eta ,k)h_{s}(-k_{x}-\xi ,-k_{y}-\eta ,k)d\xi d\eta$$

It is clear from the structure of Equations (9) and (10) that $K_{s}$ and the lateral coordinates $\left(x_{s},y_{s}\right)$ of the scatterer enter the Fourier transform of the received amplitude $B_{s}(x_{0},y_{0};k)$ only as a factor $K_{s}e^{-i[k_{x}x_{s}+k_{y}y_{s}]}$ independent of the convolution integral $I(z_{s};k,k_{x},k_{y})$. The latter is given by Equation (10), in which the axial position $z_{s}$ of *s*-th scatterer enter via the dependence of function $h_{s}(\ldots )$ on $z_{s}$.

## 4. Spectral Form of the Filtering Function to Transform the Initial Set of A-Scans $A\left(x_{0},y_{0},z_{q}\right)$ to Arbitrary Target form $A^{\left(T\right)}\left(x_{0},y_{0},z_{q}\right)$

Now, we consider how the above-derived relationships can be used to solve the following problem. Let us assume that the illuminating field has the complex-valued amplitude $U_{L}(x,y;k)$ after the illuminating-lens aperture and the acquired A-scans are $A\left(x_{0},y_{0},z_{q}\right)$. Our goal is to transform the initial 3D set of A-scans $A\left(x_{0},y_{0},z_{q}\right)$ to another (“target”) form $A^{\left(T\right)}\left(x_{0},y_{0},z_{q}\right)$. This means that the target form of scans corresponds to another field distribution $U_{L}^{\left(T\right)}(x,y;k)$ at the illuminating aperture:(11)$$U_{L}(x,y;k_{n})\to U_{L}^{\left(T\right)}(x,y;k_{n})$$

In particular, the distribution $U_{L}(x,y;k)$ may contain some aberrations at the illuminating/receiving aperture, whereas $U_{L}^{\left(T\right)}(x,y;k)$ is a desired aberration-free distribution. In other cases, the desired distribution $U_{L}^{\left(T\right)}(x,y;k)$ may correspond, for example, to refocusing to another depth.

We recall that the field distribution $U_{L}(x,y;k_{n})$ in the K-space is described by the angular spectrum $g(k_{x},k_{y},k)$ given by integral (1). Similarly, the target distribution $U_{L}^{\left(T\right)}(x,y;k_{n})$ in the spectral representation corresponds to $g^{\left(T\right)}(k_{x},k_{y},k)$, to obtain which one should use $U_{L}^{\left(T\right)}(x,y;k_{n})$ instead of $U_{L}(x,y;k_{n})$ in Equation (1). Then, in integral relationship (10) one should substitute $g(k_{x},k_{y},k)$ by $g^{\left(T\right)}(k_{x},k_{y},k)$ to obtain the target convolution function $I^{\left(T\right)}(z_{s};k,k_{x},k_{y})$.

Next, we recall that Equations (9) and (10) define the angular spectrum ${\hat{B}}_{s}(k_{x},k_{y};k)$ of the received signal from *s*-th scatterer for the initial field distribution $U_{L}(x,y;k_{n})$. Equations with the same structure as (9) and (10) also define angular spectrum ${\hat{B}}_{s}^{\left(T\right)}(k_{x},k_{y},k)$ for the target distribution $U_{L}^{\left(T\right)}(x,y;k_{n})$. Consequently, the angular spectrum ${\hat{B}}_{s}(k_{x},k_{y},k)$ can be transformed to the desired form ${\hat{B}}_{s}^{\left(T\right)}(k_{x},k_{y},k)$ via the following filtering function:(12)$$Filt(z_{s};k,k_{x},k_{y})=\frac{{\hat{B}}_{s}^{\left(T\right)}(k_{x},k_{y},k)}{{\hat{B}}_{s}(k_{x},k_{y},k)}\equiv \frac{DFT_{x_{0},y_{0}\to k_{x},k_{y}}\{B_{s}^{\left(T\right)}(x_{0},y_{0};k)\}}{DFT_{x_{0},y_{0}\to k_{x},k_{y}}\{B_{s}(x_{0},y_{0};k)\}}=\frac{I^{\left(T\right)}(z_{s};k,k_{x},k_{y})}{I(z_{s};k,k_{x},k_{y})}$$

An important feature of the filtering function (12) is that it does not depend on lateral coordinates $\left(x_{s},y_{s}\right)$ of scatterers and their scattering strengths $K_{s}$, which is due to the factorized form of Equation (9). It also worth noting that for avoiding the occasional division by nearly zero values, in the denominator of Equation (12) some regularization term α>0 should be added $I(z_{s};k,k_{x},k_{y})\to I(z_{s};k,k_{x},k_{y})+\alpha$.

Rigorously speaking, the filtering function in Equation (12) depends on the wavenumber $k=k_{n}$ of every component of the illuminating optical field. Equation (12) also indicates that the filtering depends on the scatterer depth $z=z_{s}$. However, we recall that unlike coordinate $z_{s}$ of *s*-th scatterer, which is a continuous quantity, the depth coordinate $z_{q}$ in the reconstructed OCT scans (see Equation (7)) is discreet. Thus, for transformation of OCT scans it makes sense to define the depth-dependent filtering function (12) for discreet depth points $z_{q}$.

Now, one can obtain the corrected (target) A-scan $A^{\left(T\right)}\left(x_{0},y_{0},z_{q}\right)$ by transforming the initial A-scan $A\left(x_{0},y_{0},z_{q}\right)$ given by Equation (7) to the K-space, then multiplying the so-found spectrum by the filter function (12) and then again applying the Fourier transform to return to the corrected spectrum to the spatial domain. This procedure looks as follows:(13)$$A^{\left(T\right)}\left(x_{0},y_{0},z=z_{q}\right)=DFT_{k_{x},k_{y}\to x_{0},y_{0}}\left[DFT_{k_{n}\to z_{q}}\left\{\begin{array}{l} Filt(z_{q};k_{n},k_{x},k_{y})\times \\ \times DFT_{x_{0},y_{0}\to k_{x},k_{y}}\left(DFT_{z_{q}\to k_{n}}[A(x_{0},y_{0},z_{q})]\right) \end{array}\right\}\right]$$

Now, one may recall that the spectral composition $S(k_{n})$ of the illuminating beam for OCT devices is well localized around the central wavenumber $k_{0}\approx k_{n}$. Taking this into account, Equation (13) can be significantly simplified by eliminating the inverse and forward Fourier transforms over the wavenumbers $k_{n}$. The latter can be substituted by the central wavenumber $k_{0}$. Then, the simplified version of Equation (13) takes the form:(14)$$A^{\left(T\right)}\left(x_{0},y_{0},z=z_{q}\right)\approx DFT_{k_{x},k_{y}\to x_{0},y_{0}}\left\{Filt(z_{q};k_{0},k_{x},k_{y})\times DFT_{x_{0},y_{0}\to k_{x},k_{y}}[A(x_{0},y_{0},z_{q})]\right\}$$

In comparison with Equation (13), the simplified Equation (14) strongly reduces the computational expenses with only an insignificant influence on the accuracy of the entire correction procedure.

Thus, the derived Equations (13) and (14) (together with Equation (12) for the filtering function in the k-space) describe the OCT-scan transformation to the desired form. The transformation steps can be summarized in the diagram shown in Figure 3. In the next section, we will present some examples of application of the described transformation procedure.

## 5. Examples of Transformation Procedures $A\left(x_{0},y_{0},z_{q}\right)\to A^{\left(T\right)}\left(x_{0},y_{0},z_{q}\right)$

### 5.1. Conventional Refocusing

The first example demonstrating the efficiency of the above-derived relationships is their application for obtaining a deeper insight into the widely discussed problem of digital refocusing of OCT data. For digital shifting, the focus depth using a 3D set of OCT data obtained with a Gaussian beam, the following procedure based on several spectral transformations is known [25,26]:(15)$$A_{\Delta z}\left(x,y,z\right)=\sum_{k}\exp(ikz)\cdot DFT_{k_{x},k_{y}\to x,y}\left[DFT_{x,y\to k_{x},k_{y}}\left[DFT_{z\to k}\left[A_{0}(x,y,z)\right]\right]\cdot \exp\left(-i\cdot \Delta z\cdot \frac{k_{x}^{2}+k_{y}^{2}}{4\cdot k}\right)\right]$$
where $A_{0}\left(x,y,z\right)$ is the initial field and $A_{\Delta z}\left(x,y,z\right)$ is the optical field after the numerical shift of the focal plane at a distance Δz. In paper [25], this procedure was proposed using heuristic arguments without rigorous derivation. In what follows, we clarify in detail the background approximations and demonstrate how this expression can be rigorously derived from the general Equation (12) for the filtering function and Equations (13) and (14) for spectral transformations based on this function.

Let us consider the illuminating beam $U_{beam}$ of a Gaussian form using the notations from work [11] assuming the focal depth $z=z_{0}$:(16)$$\begin{array}{l} U_{beam}\left(x,y,z;k\right)=\frac{W_{0}}{W\left(z,z_{0}\right)}\cdot \exp\left(-\frac{{\left(x-x_{0}\right)}^{2}+{\left(y-y_{0}\right)}^{2}}{W^{2}\left(z,z_{0}\right)}\right)\times \\ \times \exp\left\{-i\cdot \left[k\cdot \left(z-z_{0}\right)-\Phi \left(z,z_{0}\right)-k\frac{{\left(x-x_{0}\right)}^{2}+{\left(y-y_{0}\right)}^{2}}{2\cdot R\left(z,z_{0}\right)}\right]\right\} \end{array}$$

Here, $W_{0}$ is the radius of the focal waist at $z=z_{0}$ and the current beam width $W\left(z,z_{0}\right)$ as a function of z is given by
(17)$$W\left(z,z_{0}\right)=W_{0}\cdot [1+{\left(\frac{\lambda \cdot \left(z-z_{0}\right)}{\pi \cdot W_{0}^{2}}\right)}^{2}{]}^{1/2}$$
where λ=2π/k. Assuming that the illuminating/receiving aperture is located at z=0, $W\left(0,z_{0}\right)\equiv W_{L}$ is the initial radius of the beam at z=0. Function $\Phi \left(z,z_{0}\right)=arctg\left(\lambda \cdot \left(z-z_{0}\right)/(\pi \cdot W_{0}^{2})\right)$ describes the phase difference between the Gaussian beam and an ideally plane wave propagating the same distance $\left(z-z_{0}\right)$. This phase term does not affect the beam focusing because it is independent of lateral coordinates, so in the following consideration it may be omitted. Quantity $R\left(z,z_{0}\right)$ in Equation (16) is the current phase-front curvature
(18)$$R\left(z,z_{0}\right)=\left(z_{0}-z\right)\cdot [1+{\left(\frac{\pi \cdot W_{0}^{2}}{\lambda \cdot \left(z_{0}-z\right)}\right)}^{2}]$$
so that $R\left(0,z_{0}\right)$ is the initial curvature at the aperture. At this point, one may recall that the necessity of refocusing is important for highly focused OCT systems, for which $W_{0}\sim \lambda$ near the focus and $\pi \cdot W_{0}^{2}/\left(\lambda z_{0}\right)<<1$, so that
(19)$$R\left(0,z_{0}\right)=z_{0}[1+{\left(\pi \cdot W_{0}^{2}/\left(\lambda z_{0}\right)\right)}^{2}]\approx z_{0}$$

Similarly, in view of $\pi \cdot W_{0}^{2}/\left(\lambda z_{0}\right)<<1$, the relationship (17) between the focal radius $W_{0}$ and the beam radius $W\left(0,z_{0}\right)\equiv W_{L}$ at the aperture is also simplified
(20)$$W_{0}\approx \lambda z_{0}/(\pi W_{L})$$

Thus, omitting pre-factors (including the term with $\Phi \left(0,z_{0}\right)$) independent of lateral coordinates and, therefore, insignificant for focusing, the initial field distribution over the aperture at z=0 can be written as
(21)$$U_{L}\left(x,y,0;k\right)=C\cdot \exp\left(-\frac{{\left(x-x_{0}\right)}^{2}+{\left(y-y_{0}\right)}^{2}}{W_{L}^{2}}\right)\times \exp\left\{-i\cdot k\frac{{\left(x-x_{0}\right)}^{2}+{\left(y-y_{0}\right)}^{2}}{2\cdot z_{0}}\right\}$$

Here, pre-factor C comprises independent lateral coordinate terms that are insignificant for focusing. The first exponential factor describes the distribution of the illuminating beam amplitude over the aperture plane z=0, whereas the position of the focus corresponding to $z=z_{0}$ is determined by the wave-front curvature. For the Gaussian distribution (21) at the aperture, in the approximation that $W_{L}>>{\left(\lambda z_{0}\right)}^{1/2}$ (i.e., with the requirement that there are many Fresnel zones over the aperture), the angular spectrum $g(k_{x},k_{y},k)$ has the form
(22)$$g(k_{x},k_{y},k)=const\cdot \exp\left\{-\left(k_{x}^{2}+k_{y}^{2}\right)\frac{z_{0}}{2\cdot k}\cdot (\frac{2z_{0}}{kW_{L}^{2}}-i)\right\}$$

Now spectrum (22) can be used to find the filtering function (12). To this end, the auxiliary integral $I(z_{s};k,k_{x},k_{y})$ given by Equation (10) should be first evaluated for the initial position $z_{0}$ of the illuminating-beam focus, and then target integral $I^{\left(T\right)}(z_{s},k,k_{x},k_{y})$ should be similarly evaluated for any desired shifted focus distance $z_{0}^{\prime }=z_{0}-\Delta z$. In the general case, integral (10) can be evaluated numerically by taking the propagator function $h_{s}(\ldots )$ in the full form given by Equation (2) in combination with function g(…) given by Equation (22). However, only for comparison with previously known results, we intentionally perform this evaluation analytically using the propagator $h_{s}(\ldots )$ in the paraxial approximation instead of full Equation (2):(23)$$h_{s}(k_{x},k_{y},k)=e^{iz_{s}\sqrt{k^{2}-k_{x}^{2}-k_{y}^{2}}}\approx \exp\left\{iz_{s}\cdot (k-\frac{k_{x}^{2}+k_{y}^{2}}{2k})\right\}$$

In this approximation, the integrand in Equation (10) contains the following expression:(24)$$g(k_{x},k_{y},k)h_{s}(k_{x},k_{y},k)=const\cdot \exp\left\{-(k_{x}^{2}+k_{y}^{2})\cdot \frac{z_{0}}{2k}(\frac{2z_{0}}{kW_{L}^{2}}-i+i\frac{z_{s}}{z_{0}})\right\}\exp(iz_{s}k)$$

For further consideration, we note that the last factor in Equation (24) independent of $k_{x,y}$ can be omitted. Both the angular spectrum $g(k_{x},k_{y},k)$ given by Equation (22) and the propagator function $h_{s}(k_{x},k_{y},k)$ in Equation (24) contain phase factors with similar quadratic dependences on the lateral wave-vectors $k_{x,y}$: $\exp\left\{i\cdot z_{0}(k_{x}^{2}+k_{y}^{2})/(2\cdot k)\right\}$ and $\exp\left\{-i\cdot z_{s}\cdot (k_{x}^{2}+k_{y}^{2})/(2\cdot k)\right\}$, for which integral (10) can be found analytically. The resultant expression obtained from Equation (10) for integral I(…) (i.e., for the initial focus distance $z=z_{0}-z_{o}^{\prime }$ and scatterer depth $z_{s}$) contains the following factors
(25)$$I(\ldots )\propto \exp\left\{-(k_{x}^{2}+k_{y}^{2})\frac{z_{0}^{2}}{2k^{2}W_{L}^{2}}\right\}\exp\left\{i\cdot (k_{x}^{2}+k_{y}^{2})\frac{z_{0}-z_{s}}{4k}\right\}$$

In the target integral $I^{\left(T\right)}(\ldots )$ we keep the same scatterer position $z_{s}$ (that is inherited from Equation (23) for $h_{s}(\ldots )$), but require that the new position $z_{0}^{\prime }$ of the focal plane is shifted by $\Delta z=z_{0}-z_{0}^{\prime }$ and assume that the beam radius and the aperture $W_{L}^{\prime }$ is not necessarily equal to the initial $W_{L}$. Thus, $I^{\left(T\right)}(\ldots )$ takes the form functionally very similar to Equation (25)
(26)$$I^{\left(T\right)}(\ldots )\propto \exp\left\{-(k_{x}^{2}+k_{y}^{2})\frac{{z^{\prime }}_{0}^{2}}{2k^{2}{W^{\prime }}_{L}^{2}}\right\}\exp\left\{i\cdot (k_{x}^{2}+k_{y}^{2})\cdot \frac{z_{0}^{\prime }-z_{s}}{4k}\right\}$$

Finally, the filtering function (12), equal to the ratio *I*^(*T*)^(…)/*I*(…) takes the form
(27)$$Filt(\ldots )=\frac{I^{\left(T\right)}(z_{s};k,k_{x},k_{y})}{I(z_{s};k,k_{x},k_{y})}=\exp\left\{-i\cdot (k_{x}^{2}+k_{y}^{2})\frac{\Delta z}{4\cdot k}\right\}\exp\left\{-\frac{\left(k_{x}^{2}+k_{y}^{2}\right)}{2k^{2}}(\frac{{z^{\prime }}_{0}^{2}}{{W^{\prime }}_{L}^{2}}-\frac{z_{0}^{2}}{W_{L}^{2}})\right\}$$

It is clear that the refocusing factor in the earlier introduced Equation (15) differs from Equation (27) in which, besides the first phase exponential factor, an additional second exponential factor is present. This second exponential factor in Equation (27) disappears if
(28)$${z^{\prime }}_{0}^{2}/{W^{\prime }}_{L}^{2}=z_{0}^{2}/W_{L}^{2}$$

Then, Equation (27) reduces to the same phase-only correction (in the spectral domain) as in Equation (15). However, this actually means that in the spatial domain, Equation (15) corresponds to the *variable beam radius* described by Equation (28). Due to this correction of the beam radius dependent on the refocusing depth, the focus waist after refocusing, based on Equation (15), keeps exactly the same radius as in the physical focus. This is clear from the relationship (20) between $W_{0}$ and $W_{L}$. In contrast, if the beam radius in Equation (27) is invariable at the aperture, $W_{L}=W_{L}^{\prime }$, then after refocusing, the focus-waist radius is either smaller (for $z_{0}^{\prime }<z_{0}$) or larger (for $z_{0}^{\prime }>z_{0}$) than in the physical focus at $z=z_{0}$. We emphasize that the general Equation (12) for the filtering function is reduced to analytical Equation (27) and to previously known Equation (15), assuming the paraxial approximation and under the assumption that the curvature radius at the aperture is approximated as $R\left(0,z_{0}\right)=z_{0}[1+{\left(\pi \cdot W_{0}^{2}/\left(\lambda z_{0}\right)\right)}^{2}]\approx z_{0}$. Note finally that, by analogy with general Equation (14), with good accuracy Equation (15) can also be represented in the simplified form
(29)$$A_{\Delta z}\left(x,y,z\right)=DFT_{k_{x},k_{y}\to x,y}\left[\exp\left(-i\cdot \Delta z\cdot \frac{k_{x}^{2}+k_{y}^{2}}{4\cdot k}\right)DFT_{x,y\to k_{x},k_{y}}\left[A_{0}(x,y,z)\right]\right]$$

This simplified form requires a much smaller number of calculations than Equation (15), and is especially computationally efficient.

Figure 4a demonstrates a simulated OCT B-scan representing a vertical chain of seven point-like scatterers imaged by a highly focused Gaussian beam, for which only the central scatterer located at the focus depth $z_{s}=z_{0}=252$ μm is imaged with the maximal resolution. In this example, a central wavelength of 0.85 μm is chosen as quite a typical value for OCT devices (however, for another desirable wavelength, the results may be easily recalculated). Details of the usage of model Equations (5)–(7) for scan generation can be found in [19]. Notice that these equations also readily allow one to generate OCT scans with a high density of scatterers. Examples of the latter can be found in [19], but in such scans, regions near and far from the focus depth visually do not clearly differ. Therefore, in Figure 4 and subsequent examples, we intentionally use only several distinct scatterers to clearly show the effects of beam transformations. Figure 4a clearly demonstrates that out of focus, the point-spread function becomes broader and gets a bit curved. The lateral size of the simulated scan in Figure 4 is chosen to be ~3–4 times greater than the illuminating-beam diameter in the out-of-focus regions to reduce the influence of boundaries on the results of the digital Fourier transformations required for refocusing. A further increase in simulated region size technically is not a problem, but this does not make much sense because in the too large field of view, the sizes of the initial and refocused scatterer images would be poorly seen for clear comparison.

Figure 4b shows the result of digital refocusing based on Equation (15) which actually performs depth-dependent adjusting of the optical-beam radius at the illuminating aperture and thus enables the depth-independent focus radius. Figure 4c,d show the results of refocusing with the use of the proposed filtering function (12) and Equation (14) for constructing refocused images under two assumptions: first, assuming the invariable beam width $W_{L}$ at the illuminating aperture (Figure 4c), and then invariable focus radius $W_{0}$, but correspondingly corrected $W_{L}$ for the current focus depth according to Equation (20) (Figure 4d).

To conclude this discussion, it can be mentioned that in Equations (15) and (29) the shift Δz of the focus looks like a fixed value. However, to obtain refocusing of image $A_{\Delta z}\left(x,y,z\right)$ at all depths $z=z_{p}$ (where p=1…N), one should recall that for pixels with different coordinate z, the focus shift Δz from the initial position $z_{0}$ should be different, $\Delta z=z_{0}-z$. Bearing this condition in mind, for simultaneous refocusing of pixels at all depths z, Equation (29) can be rewritten as
(30)$$A_{\Delta z}\left(x,y,z\right)=DFT_{k_{x},k_{y}\to x,y}\left[\exp\left(-i\cdot (z_{0}-z)\cdot \frac{k_{x}^{2}+k_{y}^{2}}{4\cdot k}\right)DFT_{x,y\to k_{x},k_{y}}\left[A_{0}(x,y,z)\right]\right]$$

We point out that the matrix expression (30) performs a one-step transformation of the initial 3D set of pixelated OCT data $A_{0}(x,y,z)$ into another 3D array $A_{\Delta z}\left(x,y,z\right)$. As a result, *pixels at all depths*
$z=z_{p}$
*become refocused* without the need of additional operations (such as auxiliary transforms between the spatial and z-frequency domains, etc.). Actually, the refocused images in Figure 4 were obtained in such a way.

### 5.2. A Non-Gaussian and Non-Paraxial Example of Using the Spectral Filtering Function

In Section 5.1, we performed a comparison of OCT-image refocusing based on an earlier known correcting function of the phases of angular spectral components given by Equation (15), with the derived generalized spectral filtering function (12). For a clearer comparison, we evaluated the latter function using the conventional paraxial approximation and Gaussian beams, the properties of which are widely known. These approximations made it possible to analytically derive approximate forms of the auxiliary integrals (10) for the initial and target beams and thus reveal the connection between Equations (10)–(12) and the earlier known results.

However, we emphasize that numerically performed transformations based on the filtering function given by Equations (10)–(12) require neither paraxial approximation nor Gaussian beams. To illustrate the general nature of Equations (10)–(12), in this Section, we will use the full non-paraxial form of the propagator function (2) and intentionally demonstrate a bit of an exotic example of obtaining an OCT image for a non-diverging Bessel illuminating beam starting from an initial image acquired by a highly focused non-Gaussian beam. Namely, we start with a beam having a Lorentzian amplitude profile and spherical phase distribution over the aperture, for which amplitudes of spectral components $k_{x,y}$ decrease significantly slower than for a Gaussian beam, namely, $\propto \exp(-|k_{x,y}|)$ instead of $\propto \exp(-k_{x,y}^{2})$. For a Bessel beam, its amplitude profile at the aperture should be Gaussian, but the phase distribution should correspond to passing through an axicon lens [14,15] spectral filtering function (12) for such initial and target profiles is found digitally, using the numerical evaluation of auxiliary integrals (10) for the Lorentzian and Bessel-beam profiles. Figure 5 corresponds to the same set of scatterers as in Figure 4 and shows the vertical and horizontal OCT images corresponding to the initial Lorentzian beam and resultant Bessel beam. The intensities of scatterer images in Figure 5(a-1) are intentionally equalized to better show the out-of-focus scatterers. It is known that Bessel beams have noticeable sidelobes, but the central maximum well keeps its size independent of the depth and is comparable with the focal spot of the initial strongly focused beam. For clarity, this example is simulated in the absence of noise. The influence of noise on the results of such transformations and the necessity of the corresponding regularization procedures will be discussed in the next section.

### 5.3. Super-Refocusing

In Section 5.1, we demonstrated the use of the derived spectral filtering function (12) for digital refocusing of an initial 3D set of OCT data. The analysis based on Equation (12) revealed that, for ensuring depth-independent lateral resolution in the refocused OCT data, the same as in the physical focus, it was necessary to manipulate both phase curvature and the beam width at the aperture. This combined phase-amplitude manipulation is not obvious from the form of filtering function given by Equation (15), which looks like a phase-only correction in the spectral domain. The use of the proposed generalized filtering function (12) in Section 5.1 made it possible to demonstrate that the choice of equal initial and target beam radii $W_{L}=W_{L}^{\prime }$ at the aperture resulted in a variable focal radius $W_{0}^{\prime }$, so that for $z_{0}^{\prime }<z_{0}$, the $W_{0}^{\prime }$ is somewhat smaller than $W_{0}$ in the physical focus.

Now, like in Section 5.1, we consider the most interesting for practice Gaussian beams and demonstrate how the filtering function (12) can be used for phase-amplitude manipulation with the initial OCT data to enable “super-refocusing”. Possibilities of increasing resolution in optics in general have been analyzed for a long time (e.g., review [27]). Specifically for OCT, these issues were recently discussed in [28] using both theoretical argumentation based on the information capacity of optical images (similarly to [27]) and even some experimental examples of resolution enhancement. Such “super-refocusing” does not mean that the diffraction limit is overcome, nevertheless, proper modification of the phase-amplitude beam profile can enable a significantly smaller target focus radius than in the physical focus, $W_{0}^{T}<W_{0}$. In what follows, we use the derived generalized filtering function (12) to obtain such resolution enhancement and demonstrate the limiting role of measurement noises on the efficiency of such super-refocusing.

Recalling the relationship $W_{0}\approx \lambda z_{0}/(\pi W_{L})$ following from Equation (17) and the filtering function (27) derived for conventional refocusing of Gaussian beams in Section 5.1, it is clear that to enable invariable $W_{0}^{T}<W_{0}$, one should choose in Equation (27) invariable ratio $z_{0}^{\prime }/W_{L}^{\prime }<z_{0}/W_{L}$ for every refocusing depth $z_{0}^{\prime }$, and simultaneously the beam width $W_{L}^{\prime }$ should be chosen broader than the initial beam radius.

An example of the application of such procedures for super-refocusing is presented in Figure 6. In this example, the simulation is made in the presence of noise in the initial image, assuming a typical OCT SNR~30 dB (Figure 6(a-1,a-2)). The initial Gaussian illuminating beam is assumed to be weakly focused (with a focus radius of $W_{0}=12.6$ μm and the central wavelength λ=0.85 μm is the same as in the previous examples). Now, instead of a vertical chain of individual scatterers, we consider a similar chain of pairs of point-like scatterers laterally separated by 12 μm. In the image formed by such a weakly focused beam, the scatterers in the pairs are not resolved (see the vertical B-scan in Figure 6(a-1) and the en-face image of one of the pairs at z=480 μm in the inset).

The demonstrated increase in resolution is attained due to the effective increase in the illuminating beam width at the aperture, or, in other words, due to applying an amplitude mask, which causes flattening of the beam profile. However, flattening of the beam is performed at the expense of suppression of its amplitude near the beam axis. This effect certainly reduces the amplitude of the retained signals from the scatterers and decreases the resultant SNR. Nevertheless, as demonstrated in Figure 6(a-1,a-2), for quite a realistic initial SNR of the order of 25–30 dB, the SNR remains at an acceptable level after super-refocusing (see Figure 6(c-1,c-2)). Thus, such super-refocusing can be applied to significantly improve the imaging resolution by re-processing already acquired datasets made with low-resolution OCT setups if the initial SNR is not too bad.

In terms of the angular spectrum of the beam, this improvement in resolution is due to digitally performed enhancing of the out-of-center angular spectral components. These side spectral components initially have smaller amplitudes and, therefore, in real measurement, are fairly noisy. Consequently, enhancement of these components may significantly increase the noise in the super-refocused image. Thus, to prevent unacceptable noise increases, some regularization procedures should be used when performing the signal transformation. Although quite sophisticated forms of regularization are used in some works (e.g., [29]), their detailed discussion is far beyond the scope of this paper. In view of this, we limit ourselves to the simple regularization method mentioned after the expression (12) for the spectral filter. Namely, to avoid division by near-zero values, we add a small regularization parameter α>0 to the denominator in Equation (12). This parameter strongly affects the resultant noise level in the transformed OCT image.

To illustrate this statement, Figure 7 shows the simulated super-refocused profiles of the OCT signal through the pair of scatterers located at z=480 µm, similarly to Figure 6(c-2), but using various values of α. The upper two rows Figure 7(a-1–a-5,b-1–b-5) show the simulated results in the absence of any measurement noises. All profiles are normalized to the peak amplitudes corresponding to the smallest $\alpha =10^{-6}$, these amplitudes are taken as unity value (0 dB in Figure 7(a-1)). Figure 7(a-1–a-5)) demonstrate that a gradual increase in α by 6 orders of magnitude leads to a gradual decrease in the maxima (up to a 10 dB decrease for α=1 in Figure 7(a-5), in comparison with the initial 0 dB for $\alpha =10^{-6}$ in Figure 7(a-1)). At the same time, the minimum in the center between the scatterers becomes less deep. Simultaneously, noise-like sidelobes around the scatterer images arise with increasing α.

The next two rows, Figure 7(c1–c5,d1–d5), show how the super-refocused images and their profiles look for the same set of α-values in the presence of moderate noise, the same as in the image before refocusing in Figure 6(a-1,a-2). It is clear that for too small $\alpha =10^{-6}$ and $\alpha =10^{-4}$, the super-refocused images of scatterers in Figure 7(c-1,c-2,d-1,d-2) are completely corrupted by noises. The level of the latter becomes much higher than in the initial images shown in Figure 6(a-1,a-2) before super-refocising.

For $\alpha =10^{-3}$ (Figure 7(c-3,d-3)) the scatterers after super-refocusing become visible, but still look rather noisy. Parameter $\alpha =10^{-2}$ (Figure 7(c-4,d-4)) gives the best results with a reasonable compromise between the reduction in noise and the depth of the inter-scatterer minimum. This agrees with Tikhonov’s theory of regularization [30], according to which the optimal value of α should be of the order of the inverse SNR. Further, Figure 7(c-5,d-5) show that for too-big regularization parameter α=1, the noises are further suppressed; however, the resolution degrades. Namely, the peaks corresponding to individual scatters become broader, whereas the minimum between the scatterers becomes shallower. Figure 6 and Figure 7 show that the improvement of resolution by a factor of 4 in this example leads to ~2–2.5 times a decrease in SNR for the optimal regulation parameter $\alpha =10^{-2}$. Thus, the improvement of resolution after super-refocusing is obtained at the expense of decrease in SNR in the resultant image. If the initial SNR is not too small, after duly chosen regularization degree corresponding to the initial noise, the resultant SNR and the resolution of the image after super-refocusing may be still acceptable as in Figure 7(c-4,d-4).

Although such super-refocusing improves the initial lateral resolution, it cannot overcome the diffraction limit, even if the initial image has a very high SNR. Furthermore, the digital enhancement of the effective numerical aperture of the illuminating beam may be limited by the physical lens aperture; the latter also limits the attainable effective value of the numerical aperture and, thus, the possible degree of super-refocusing.

### 5.4. Correction of Aberrations at the Illuminating/Receiving Aperture

Another problem for which the developed K-space framework can be applied is the correction of aberrations at the illuminating/receiving aperture. Here, we consider the situation in which the initial field distribution over the aperture $U_{L}(x,y,k_{n})$ is characterized by the presence of aberrations Q(x,y). Thus, the target aberration-free distribution $U_{L}^{\left(T\right)}(x,y,k_{n})$ is related to the initial distribution as follows:(31)$$U_{L}(x,y,k_{n})=U_{L}^{\left(T\right)}(x,y,k_{n})Q(x,y)$$

The estimation of unknown forms of aberrations represents a non-trivial problem, and various aspects of this problem were discussed in the literature (e.g., [31,32,33,34,35,36,37]). However, for the moment, we focus on the aberration compensation itself, assuming that the aberration-function Q(x,y) is known. At this point, it is important to recall that, because of the double passing through the illuminating/receiving aperture, the acquired OCT signal is proportional to the square of the field incident at scatterers (see Equation (6)). For this reason, the aberration-induced distortions are nonlinearly mixed into the received OCT signal. In view of this, the compensation procedure is less trivial than in the one-pass situation, in which, for example, the influence of phase-only distortion $Q(x,y)=\exp(i\Phi_{dist})$ can readily be corrected by applying the phase-conjugated function $\propto \exp(-i\Phi_{dist})$. But for the considered double-pass case, neither the single phase correction $\propto \exp(-i\Phi_{dist})$ nor the double phase correction $\propto \exp(-i2\Phi_{dist})$ are sufficient. However, similarly to the refocusing problems, we can use the scheme presented in Figure 3 for correction of aberrations described by the function Q(x,y) in Equation (31). Thus, the filtering function (12) can be found as the ratio of integrals (10) calculated for the desired target distribution $U_{L}^{\left(T\right)}(x,y,k_{n})$ and the distorted distribution (31) containing the given aberration function Q(x,y). The aberrations are often represented as a superposition of Zernike polynomials [22,23,24,25,26,27,28,29,30,31,32,33,34,38,39], so in what follows, we also use a Zernike polynomial to demonstrate the procedure of aberration correction using the scheme in Figure 3. To consider a fairly complex, axially asymmetric aberration type, we choose the Zernike polynomial $Z_{3}^{1}(\rho ,\phi )$ (“coma”), so that $Q(x,y)=\exp[-iA_{0}Z_{3}^{1}(\rho ,\phi )]$, where $A_{0}$ is the aberration amplitude (in the examples shown below, it was 60 rad.)

The upper row in Figure 8 shows examples of OCT images obtained in the presence of aberrations for the same focused illuminating beam and the same vertical chain of scatterers as in Figure 4a. It is clear that conventional refocusing based on Equation (15) (the middle row in Figure 8) makes the images of scatterers noticeably closer to the image in the physical focus. However, the so-refocused images still retain pronounced aberration-induced distortions of the point-spread function, as well as the lateral shift in the positions of scatterer images. The lowest row in Figure 8 (panels (a3)–(c3)) shows the results of refocusing combined with the compensation of the aberration function $Q(x,y)=\exp[-iAZ_{3}^{1}(\rho ,\phi )]$ based on the spectral filtering described by Equations (12)–(14). It is clear that the last row demonstrates both nearly perfect refocusing and compensation of the aberrations (the tiny residual distortions of <1% are related to the finite discretization step in the numerical evaluation of exact analytical expressions).

## 6. Discussion and Conclusions

In this paper, in the development of a series of earlier works related to the use of angular-spectrum approach to the description of OCT-image formation, we illustrate some possibilities of transformation of OCT data opened by this powerful approach. In the general sense, this approach develops the ideas of Fourier optics formulated several decades ago (see, e.g., textbooks [40,41]). For the realization of these principles, the progress in the development of powerful numerical packages (such as Matlab, Mathematica, etc.) opened rather convenient possibilities.

The main result of this paper is the derivation of a general form of the filtering function that enables various transformations of 3D sets of OCT data. The developed rather general and rigorous formulation of spectral-filtering procedures is summarized in Figure 3 and mathematically presented by Equations (10)–(14). To the best of our knowledge, they have not been derived before. The derived expressions in some special cases can be evaluated analytically. In more general cases, they are intended for numerical evaluation, for the realization of which such conventional assumptions as the paraxial approximation or the necessity of Gaussian beams are not required. The formulated approach, in particular, made it possible to clearly reveal conditions implicitly assumed in the conventionally discussed refocusing (compare the filtering functions in Equations (15) and (27)). We demonstrated that although this refocusing looks like the application of a phase-correcting factor in the spectral domain, actually this procedure corresponds to the digital correction of both phase and amplitude to shape the illuminating beam. Without such a simultaneous correction, the lateral size of the digitally refocused point-spread functions (i.e., the lateral resolution after refocusing) becomes dependent on the depth and may be either somewhat greater or smaller than in the initial focus (see Figure 4). To demonstrate the generality of the derived spectral filtering function (12), we presented an example of the transformation of a highly focused beam with an initial Lorentzian profile into a narrow Bessel beam.

Next, besides conventional refocusing, we demonstrated that the filtering function (12) allows one to simultaneously manipulate the phase and amplitude shape of the illuminating beam for “super-refocusing”. Certainly, this does not yet mean overcoming the diffraction limit, but the lateral resolution after such super-refocusing may become several times higher than in the initial physical focus. The ultimate resolution attainable using such super-refocusing, similarly to the physical-focus radius, is limited by the well-known diffraction limit. In real-life situations, the attainable resolution is additionally affected by the level of noise, which may be the main limiting factor in super-refocusing rather than the diffraction limit. However, for quite a realistic OCT initial signal-to-noise ratio ~25–30 dB, it is still possible to obtain an increase in the lateral resolution ~several times, as illustrated in Figure 6 and Figure 7.

Generally, the same procedures of digital filtering in the K-space may be used for other transformations of OCT signals, in particular, for compensation of aberrations as discussed in Section 5.3. Concerning other possibilities for analysis of OCT-signals and studies of various distorting factors, it was pointed out in [19,21] that the K-space description of OCT-scan formation makes it possible to simulate dispersion effects.

The procedures for forming OCT scans described here and in [19] enable a flexible and computationally efficient framework for realistic simulations of OCT signals in various situations, for which direct experimental tests are too difficult/expensive. All examples demonstrated in the manuscript required a few seconds of computation for the initial generation and transformations. Even for rather heavy simulations of forward problems, such as 3D data volumes containing large amounts of scatterers, computations can be made reasonably rapidly, although the required time increases proportionally with the number of scatterers. For example, with a volume of 128 × 128 × 128 pixels, corresponding to 192 × 192 microns laterally and 512 microns in depth with 250,000 scatterers (which imitate the density of biological cells ~5–6 microns in size), the simulations required 59 min using 12 cores of the AMD Ryzen 9 3900X CPU. However, for subsequent transformations of such an image, the computation time *does not depend* on the number of scatterers and requires only a few seconds, like in the above-presented examples with several scatterers. Numerous examples of problems in which such simulations can be used are mentioned in [19,21]. In the latter work, specific examples related to studying non-Gaussian beams or such modalities as OCT-elastography and OCT-angiography are given. Overall, the K-space-based approach opens rather diverse possibilities for numerical modeling of OCT data and their digital transformations, as well as for deep analysis of OCT-signal features for usage in biomedical imaging and other applications. We also emphasize that the utilization of such simulated (but fairly realistic) OCT data opens unique possibilities for the development and testing of various signal-processing methods in fully controllable and flexibly variable conditions, which may be very difficult/expensive, or even impossible in real physical experiments.

## Figures and Tables

**Figure 1 sensors-24-02931-f001:**
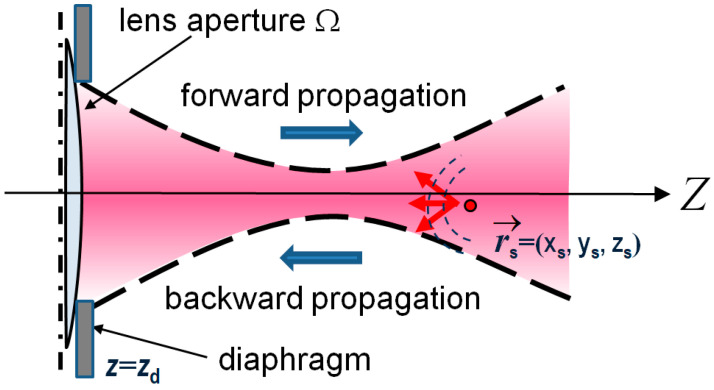
Schematically shown geometry of the illuminating beam (not necessarily focused like in the figure) with the lateral coordinates $\left(x_{0},y_{0}\right)$ of the axis directed to the tissue depth along the *z*-axis of the coordinate system. Coordinates $\left(x_{s},y_{s},z_{s}\right)$ characterize positions of scatterers.

**Figure 2 sensors-24-02931-f002:**
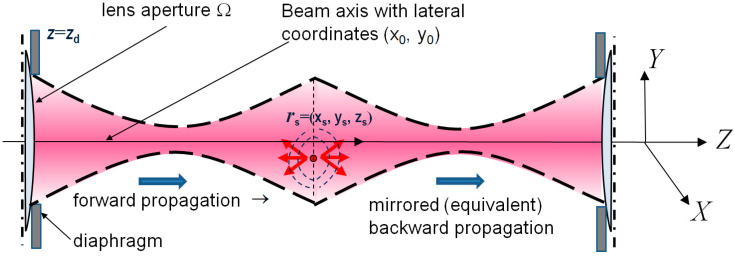
Schematic elucidation of the fact that the forward propagation of the illuminating light and back-scattered signal propagation can be described in a symmetrical manner leading to Equation (6). Notice that such an equivalent scheme should be represented for every particular scatterer depth $z_{s}$.

**Figure 3 sensors-24-02931-f003:**
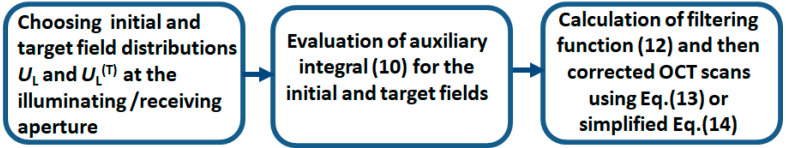
Diagram showing main steps of transformation of the complex valued OCT data obtained for the initial field distribution $U_{L}(x,y;k_{n})$ over the illuminating/receiving aperture to the desired form corresponding to a target field distribution $U_{L}^{\left(T\right)}(x,y;k_{n})$ over the aperture.

**Figure 4 sensors-24-02931-f004:**
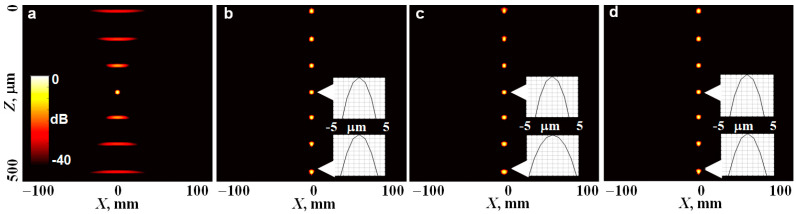
Comparison of differently performed refocusing of a B-scan for a vertical chain of sub-resolution scatterers. Panel (**a**) is the initial B-scan with a highly focused beam simulated using Equation (7) for each single A-scan. (**b**) is the result of refocusing based on Equation (15) that looks as a phase-only correction but implicitly implies variable width of the beam at the aperture, such that $W_{L}^{\prime }/z_{0}^{\prime }=W_{L}/z_{0}$, which yields $W_{0}^{\prime }=W_{0}$. (**c**) is the result of refocusing based on the filtering function (12) and Equation (14) in which the focus depth is shifted, but the beam radius at the aperture is kept invariable $W_{L}^{\prime }=W_{L0}$, so that the focus radius $W_{0}^{\prime }\neq const.$ (**d**) is the result again based on Equations (12) and (14), but requiring that $W_{0}^{\prime }=W_{0}$, such that condition $W_{L}^{\prime }/z_{0}^{\prime }=W_{L}/z_{0}$ was used when calculating the filtering function (12). The central wavelength is λ=0.85 µm, the initial focus depth is $z_{0}=252$ µm, and the radius of physical focus is $W_{0}=1.9$ µm.

**Figure 5 sensors-24-02931-f005:**
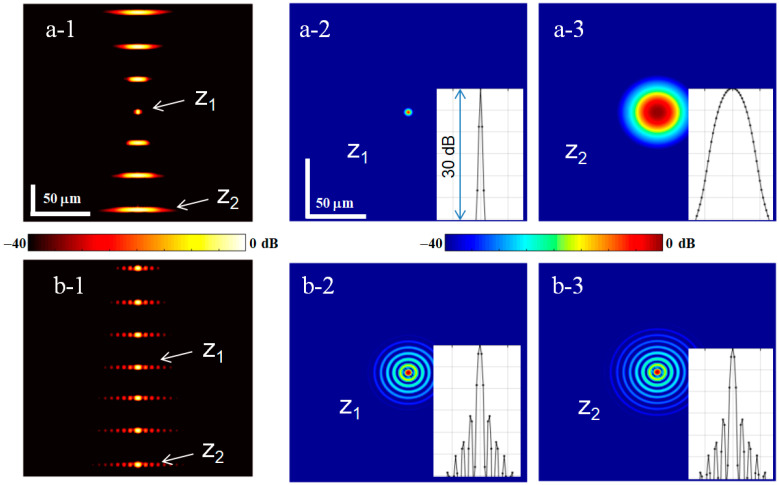
Example of transformation of non-Gaussian beams: initially strongly focused beam with a Lorentzian profile (row (**a-1**–**a-3**)) is transformed into a Bessel beam (row (**b-1**–**b-3**)). First column is for vertical B-scans, the other columns are horizontal scans through the depth $z_{1}$ of the initial focus and $z_{2}$ closer to the image bottom. The insets show the corresponding horizontal profiles.

**Figure 6 sensors-24-02931-f006:**
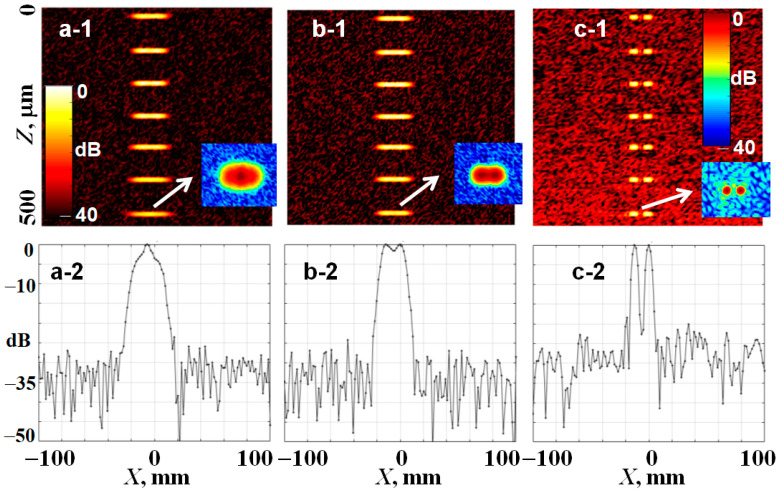
Comparison of conventional refocusing and super-refocusing for an initially weakly focused beam with $W_{0}=12.6$ μm for the same vertical positions of scatterers as in Figure 4. At each depth, a pair of scatterers laterally separated by 12 μm is located. Upper row shows color-coded B-scans with insets showing *en-face* images of scatterers located at z = 480 μm, for which the lateral profiles are shown in the lower raw. Panels (**a-1**,**a-2**) are not refocused and (**b-1**,**b-2**) show conventional refocusing similar to that in Figure 4. (**c-1**,**c-2**) show the result of super-refocusing with 4-fold increase in the lateral resolution, so that individual scatterers become clearly resolved. The noise level is especially clearly seen in the lower row.

**Figure 7 sensors-24-02931-f007:**
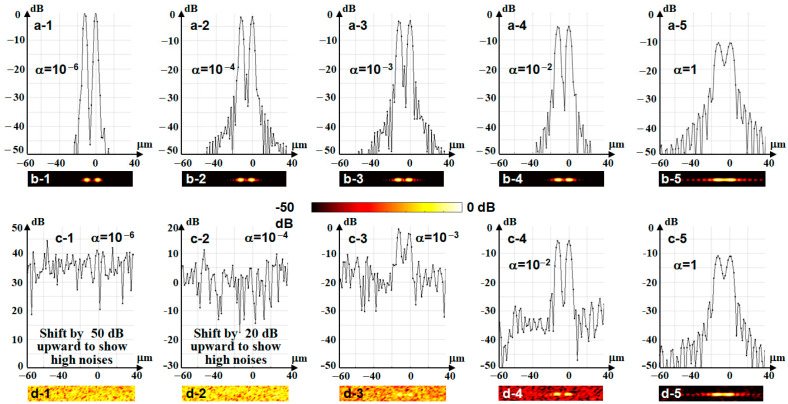
Illustration of how the regularization parameter influences the noise level in the super-refocused image with the same parameters as in Figure 5. Rows (**a-1**–**a-5**,**b-1**–**b-5**) show the “digital” noise caused by the regularization itself in the absence of other noises. Rows (**c-1**–**c-5**,**d-1**–**d-5**) correspond to the same moderate initial noise as in Figure 5 and demonstrate that the resultant noise after refocusing strongly depends on the regularization parameter.

**Figure 8 sensors-24-02931-f008:**
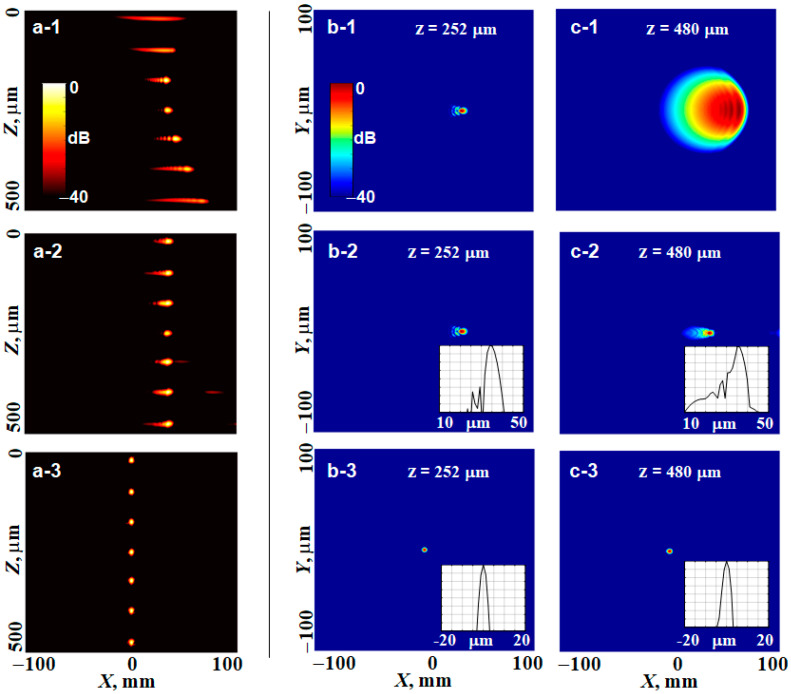
Illustration of distortions produced by the aberration in the form of Zernike polynomial $Z_{3}^{1}(\rho ,\phi )$ (**upper row**) and results of the image transformation using either conventional refocusing by Equation (15) (**middle row**) or refocusing combined with elimination of the aberrations using the spectral filter given by Equations (12) and (14) (**lower row**). Column 1 shows the in-depth B-scans, column 2 shows the *en face* image of the scatter located at the depth of physical focus z=252 μm and column 3 is the *en face* image of the scatterer located deeper at z=480 μm. The small insets show the horizontal profiles of the scatterer images along the *X*-axis.

## Data Availability

Examples of the generated numerical data can be shared upon reasonable request.

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
