# Peer review of "K-Space Approach in Optical Coherence Tomography: Rigorous Digital Transformation of Arbitrary-Shape Beams, Aberration Elimination and Super-Refocusing beyond Conventional Phase Correction Procedures"

_sensors, 2024, doi:10.3390/s24092931_

Round 1

Reviewer 1 Report (New Reviewer)

Comments and Suggestions for Authors

The article titled "K-space approach in optical coherence tomography: rigorous digital transformation of arbitrary-shape beams aberration elimination and super-refocusing beyond conventional phase correction procedures" by Alexander L. Matveyev, Lev A. Matveev, Grigory V. Gelikonov, and Vladimir Y. Zaitsev presents an innovative approach in the field of optical coherence tomography (OCT). The study introduces a K-space model for OCT-scan formation, emphasizing the digital transformation of OCT data for aberration correction and enhanced focusing beyond traditional methods.

The abstract indicates the study's main contribution as the derivation of a filtering function that enables digital transformation of OCT data to target datasets, utilizing both phase and amplitude transformations. This approach, distinct from the usual phase-only corrections, allows for the digital correction of arbitrary aberrations and non-Gaussian beam transformations, as well as a technique for "super-refocusing" that improves lateral resolution beyond the physical focus limitations, albeit with a trade-off in signal-to-noise ratio (SNR).

Given the complexity and technical depth of the article, the review will focus on evaluating the novelty, methodology, clarity, and potential impact of the research. We will proceed with a detailed analysis of the document to prepare a comprehensive review report.

The article presents a novel K-space approach for Optical Coherence Tomography (OCT) focusing on the digital transformation of arbitrary-shape beam aberrations and super-refocusing beyond traditional phase correction methods. This approach offers a comprehensive model for OCT-scan formation, utilizing both phase and amplitude transformations for digital data processing. The main outcomes include the development of a filtering function for digital aberration correction, the capability to handle non-Gaussian beam transformations, and a "super-refocusing" technique that enhances lateral resolution without surpassing the diffraction limit. The study's implications could significantly impact the OCT field by improving image quality and resolution through advanced digital processing techniques.

Author Response

We thank Referee#1 for the so high estimate of our results. According to suggestions of all referees, we introduced and highlighted in the revised manuscript some modifications which should additionally clarify some important points for readers.

Reviewer 2 Report (New Reviewer)

Comments and Suggestions for Authors

Reviewer Comments

Manuscript Title: K-space approach in optical coherence tomography: rigorous digital transformation of arbitrary-shape beams, aberration elimination and super-refocusing beyond conventional phase correction procedures

Manuscript Number: sensors-2886186

The manuscript under review is devoted to studying the usage of the proposed K-space description to analytically derive a filtering function that allows one to digitally transform the initial 3D set of complex-valued OCT data in a desired (target) data set of a rather general form. Unlike conventionally discussed phase-only transformations, The proposed filtering procedures utilize both phase and amplitude transformations.

The manuscript contains new and significant. The abstract clearly and accurately describes the content of the article. The literature review part contains distinct and rich references. The paper is nicely written and can be accepted but first, it should be improved. I have these comments:

1- Authors define the majority of abbreviations through paragraphs however some abbreviations are still not defined for example OCT in the abstract. Please verify all abbreviations.

2- In Figures 1 & 2, please indicate the beam propagation direction.

3- In Figure 4, the authors must explain the choice of the x coordinate between -100 to 100 microns.

4- In addition, in Figure 4, why did you choose the value 0.85 microns for the central wavelength? Please explain in the paper.

5- For the comparison of conventional refocusing and super-refocusing why you chose 12.6 microns for wavelength?

6- In Figure 7, how do you determine the optimal regulation parameter α.

Finally, I recommend the paper for publication after resolving the minor comments.

Author Response

General comment: The manuscript under review is devoted to studying the usage of the proposed K-space description to analytically derive a filtering function that allows one to digitally transform the initial 3D set of complex-valued OCT data in a desired (target) data set of a rather general form. Unlike conventionally discussed phase-only transformations, The proposed filtering procedures utilize both phase and amplitude transformations.

The manuscript contains new and significant. The abstract clearly and accurately describes the content of the article. The literature review part contains distinct and rich references. The paper is nicely written and can be accepted but first, it should be improved. I have these comments:

Authors: We thank the referee for the so positive summary of our results. In what follows we present our point-by-point answers to specific questions/suggestions.

Referee

1)- Authors define the majority of abbreviations through paragraphs however some abbreviations are still not defined for example OCT in the abstract. Please verify all abbreviations.

Authors: We thank the referee for pointing to us that the abbreviation OCT was used in the Abstract without definition, as well as the abbreviation “SNR” in the Introduction. This issue is corrected.

2) - In Figures 1 & 2, please indicate the beam propagation direction. 

Authors: We thank the reviewer for this suggestion. Now in Figs. 1 and 2 the directions of forward and backward propagation are explicitly indicated for convenience of the readers.

3) - In Figure 4, the authors must explain the choice of the x coordinate between -100 to 100 microns.

Authors: In Fig.  4 we perform spectral transformations of 3D OCT data corresponding in spatial domain to a rectangular parallelepiped, in the center of which the illuminating beam oriented along z-direction is placed. According to the well known properties of Fourier transforms within regions of limited sizes, the lateral directions of this parallelepiped containing the illuminating beam should exceed its diameter at least several times to exclude possible distortions of the results of Fourier transforms because of boundary effects. It is clear from Fig. 4a that in the out-of-focus regions, the effective diameter of the beam amounts to approximately 50-60 microns. Thus, we have chosen the lateral size of the region, over which the Fourier transformation is performed, equal to 200 microns (from -100 to +100 microns). This size is ~3.5-4 times greater than the effective beam diameter, which is sufficient to significantly suppress possible boundary-related distortions of the transformed fields. Further increase in the size of the simulated region is possible, but it does not make much sense because the boundary effects are already efficiently suppressed. Furthermore, for a much larger lateral size of panels in Fig. 4, the sizes of the initial and refocused images of scatterers would be too small for clear comparison. On the contrary, taking a significantly smaller region closer to the beam diameter in the defocused regions (say, from -30 to +30 pixels) could already lead to appearance of distortions related to the influence of boundaries. Thus, the chosen region size from -100 px to +100 px is a reasonable compromise.

We added such a remark in the text near Fig. 4.

4) - In addition, in Figure 4, why did you choose the value 0.85 microns for the central wavelength? Please explain in the paper.

Authors: In optical coherence tomography usually the illuminating-source wavelength is chosen in the near infrared range, typical from 0.8 to 1.3 microns. In practice, a specific wavelength is chosen taking into account spectral transparency windows of real biological tissues. We took a wavelength of 0.85 um since it is used in some real devices, but if necessary the model parameters may easily be scaled for another wavelength. Actually, for numerical demonstrations of the efficiency of the proposed approach, it is quite sufficient to choose any realistic wavelength of the order of ~1 um.

In the text we added a remark that 0.85 micron is quite a typical central wavelength for OCT devices.

5) - For the comparison of conventional refocusing and super-refocusing why you chose 12.6 microns for wavelength?

Authors: We have checked the content of the section describing super-refocusing and found that the quantity close 12 microns is mentioned twice: this is the initial focus radius W0=12.6um and 12 um is the distance between the small scattering particles. We now added a remarks that in this example we use “the central wavelength lambda=0.85 um as in the previous examples” exclude any confusion for the readers.

6) - In Figure 7, how do you determine the optimal regulation parameter α.

Authors: We added a new reference to the monograph by Tikhovonov&Arsenev (now ref.[30]), in which the principles of Tikhonov’s regularization of ill-posed problems are described, including the choice of the optimal regularization parameter. Now it is written in the manuscript that the optimal value alpha~10^(-2) “agrees with Tikhonov’s theory of regularization [30], according to which the optimal value of  should be of the order of the inverse SNR.”

Reviewer 3 Report (New Reviewer)

Comments and Suggestions for Authors

In the manuscript (Sensors-2886186), the authors proposed an OCT model based on the angular spectrum theorem. This model allows a digital filtering procedure supporting both phase- and amplitude- transformations. The derivation of the formulas is solid, and my only concern is whether it has practical value. 

In actual OCT samples, especially biological samples, scattering points' spatial location and reflectivity can be complicated. Although the authors concluded through simulation that their proposed digital filter could improve the resolution of OCT images, I wonder whether this conclusion can be transferred to an actual test. According to Figure 6, digital filters cause extra degradation in the SNR of images. Does this produce errors like ringing, ghost, or other fade characteristics? I hope the authors can validate their theory with authentic OCT images (perhaps they can find some open-source datasets). At least the details of the angular spectrum of the "real images" in the simulation should be enriched as much as possible.

In addition, it would be desirable for the authors to quantitatively evaluate how much the proposed method's resolution is improved compared to traditional methods. Can the resolution be enhanced further through parameter optimization? If so, what is the physical limit of the optimization?

Author Response

Reviewer#3

1) In the manuscript (Sensors-2886186), the authors proposed an OCT model based on the angular spectrum theorem. This model allows a digital filtering procedure supporting both phase- and amplitude- transformations. The derivation of the formulas is solid, and my only concern is whether it has practical value. 

Authors: We thank the referee for the overall positive summary of our manuscript. We also point out that there is a vast literature related to various methods intended for digital improvement of acquired sets of OCT signals (e.g. ref.[21,22]). Such improvement may include various specific goals, for example, digital refocusing of strongly focused illuminating beams in OCT to form an “effective focus” at any desirable depth within the visualized region and enable at this depth the same lateral resolution as in the physical focus. Even long before invention of OCT these was a high interest to methods intended to additionally improve the resolution of an optical system, still without overcoming the diffraction limit of focusing, but making the focusing better than in the initial optical image (e.g. ref [27]). Specifically for OCT, the possibility of “super-refocusing” also attracted attention of researchers (e.g., ref. [28]). Another problem that attracted much attention was elimination of various aberrations. It was of evident interest to enable a high quality visualization even if the used lens would have some aberrations and the initially formed image also has some aberrations. Our motivation in this study was same as for many other researchers, including the authors of studies [21,22,25,26,27,28,34,35] and numerous other works cited therein.
       We were additionally motivated by the question whether it would be possible to formulate a unified solution to apparently quite different problems of “conventional” refocusing, “super-refocusing”, elimination of aberrations and performing other digital transformations of 3D datasets in OCT. Finally we managed to find such a rather universal formulation of the sought filtering function in the K-space. This filtering function makes it possible to transform almost arbitrary initial dataset of OCT signals to another, also almost arbitrary, target form. The reported general filtering function is derived analytically and the integrals entering this function mathematically represent some convolutions. In certain special cases (like for Gaussian beams and paraxial approximation) these convolutions can be explicitly found analytically. In more general cases they can be evaluated numerically, but also rather efficiently. We consider the derivation of these general and universal expressions as the main achievement of this study.

     In certain special cases our general expressions yield the corresponding previously known formulations. This coincidence with earlier studied special cases confirms the correctness of our new formulation. Therefore, our general expressions relate to practice in the same sense as other special formulations that were earlier considered independently for various particular cases (like digital shift of the focus depth or correction of an image obtained in the presence of a given aberration). 

2) In actual OCT samples, especially biological samples, scattering points' spatial location and reflectivity can be complicated. Although the authors concluded through simulation that their proposed digital filter could improve the resolution of OCT images, I wonder whether this conclusion can be transferred to an actual test.

Authors: We agree that indeed spatial distribution of scattering centers in real samples may be very complex. However, we emphasize that the signals from scatterers obey the superposition principle. Consideration of all possible situations is not necessary because we have proven in the course of our derivations that the final results of our transformations do not depend on lateral coordinates of a scatterer (if these scatterers are not too close to visualized-region boundaries where boundary effects may cause some image degradation). Therefore, since the scattered fields obey linear superposition, the derived filtering function correctly describes transformation of a 3D image formed by an arbitrary distribution of scattering centers. Furthermore, it is important that the number of digital operations required for such transformations does not depend on the amount of scatterers, so that for a huge amount of scatterers the same time is required for transformations as for a few scatterers considered in the paper for clarity. These facts are specially explicitly emphasized in the Discussion section of the manuscript.

3) According to Figure 6, digital filters cause extra degradation in the SNR of images. Does this produce errors like ringing, ghost, or other fade characteristics?

Authors: For obtaining super-refocusing, the effective width of the illuminating beam (effective lens aperture) should be made larger. This is achieved by digitally modifying the amplitude profile of the initial beam. Namely, the beam profile should be made effectively wider, for attaining which the beam profile should be flattened. However, this flattening is attained at the expense of some loss in the illuminating-beam intensity because the required flattening is obtained by reducing the beam intensity in the center, so that effectively the beam becomes wider. In real noisy conditions such manipulations inevitably reduce the SNR in the newly formed image. We mentioned in the manuscript that the increase in the effective noise is related to appearance of very small values in the denominator of the filtering function (12) very small numbers, so that some regularization is required to reduce the noise increase in the super-refocused image. We mentioned that according to Tikhonov’s theory of regularization the value of the regularization parameter is determined by the level of noises. In the revised manuscript we also added reference book [30], in which the regularization theory is discussed in detail. Examples given in Fig. 7 give well illustrate the influence of the regularization parameter on the SNR and give a representation about the character of noise-related degradation. For too small regularization parameter, the super-refocused images of scatterers gradually become covered by noise that may completely mask the useful image. For too large regularization parameter, the noise may be strongly suppressed, but this noise suppression is attained at the expense of reduction in the resolution. This is also clearly seen from Fig. 7 for too large alpha=10^(-1). Now it is written in the manuscript that the optimal value alpha~10^(-2) “agrees with Tikhonov’s theory of regularization [30], according to which the optimal value of  should be of the order of the inverse SNR.”

Concerning regular distortions (like ringing mentioned by the referee), Fig. 6 gives an example of “ringing” at the periphery of a digitally formed Bessel beam. However, in this case the ringing is not a method-related distortion; rather this is the manifestation of sidelobes intrinsic to Bessels beams. However, sometimes distortions may appear when a scatterer is located too close to the boundary of the imaged region, or, for example, if the transformed data set does not satisfy the Kotelnikov-Nyquist condition. We highlighted such requirements to the 3D datasets in the end of Section 2 in the manuscript and additionally commented in the reply to the next question of Referee#3.

4) I hope the authors can validate their theory with authentic OCT images (perhaps they can find some open-source datasets). At least the details of the angular spectrum of the "real images" in the simulation should be enriched as much as possible.

 Authors: The reported form of the filtering function is obtained rigorously, so that there are no doubts that its application to an adequately acquired 3D sets of OCT data should give the results such as described in the manuscript. In fact the reviewer’s question is rather general and relates to any forms of equations for digital transformation of 3D data sets (certainly, including earlier derived expressions cited in the manuscript).

For such transformations, it is impossible to use individual B-scans, even phase-sensitive ones. Such transformations require an entire 3D data set. Furthermore, even if the structural 3D image looks “nicely”, the quality of these data should satisfy rather special strict conditions to guarantee operability of transformation procedures. Namely, the entire complex-valued data set should be characterized by sufficient phase stability and the spacing of neighboring A-scans should satisfy the Kotelnikov-Nyquist condition in both lateral directions. The same requirements equally relate to all previously derived expression for “conventional” digital refocusing. For example, in the cited ref [25] by Moiseev et al LPL-2012, where “conventional refocusing” was discussed, for experimental demonstration, the authors initially had to very carefully preliminarily prepare the data set for further transformations, namely, they have to perform auxiliary (also non-trivial) procedure of “phase alignment” for all A-scans in the 3D set. Our results yield the expressions earlier used in [25] as a special case. This is discussed in detail in Section 5.1. Consequently, the experimental results of paper [25] can be considered as experimental demonstration of feasibility of digital refocusing and in such a way also confirming operability of our expressions that yield the same equations as used in [25] for this special case.

The search in internet of available 3D data sets of complex-valued data or performing special own experiments with subsequent preliminary quality verification and performing phase correction procedures is an independent and rather laborious work. This it is far beyond the scope of the present study which is already rather extensive for single research paper. The main result of our present manuscript is the derivation of novel, universal expressions for OCT-data transformation, as well as demonstration of some examples of their applications of simulated OCT data with highly controllable parameters. However, in future we also plan to find a possibility for own experimental demonstrations. We also emphasize that utilization of simulated (but realistic) OCT data opens unique possibilities for the development and testing of various signal-processing methods in fully controllable and flexibly variable conditions, which is very difficult/expensive or even impossible is real physical experiments.

We added and highlighted a similar remark in the manuscript in the end of Discussion.

5) In addition, it would be desirable for the authors to quantitatively evaluate how much the proposed method's resolution is improved compared to traditional methods. Can the resolution be enhanced further through parameter optimization? If so, what is the physical limit of the optimization?

Authors:  Considering the “super-refocusing”, we already pointed out above in item 3 that we specially considered in the manuscript the degrading role of measurement noises in Fig. 7 for a particular instructive example (by comparing noiseless and noisy conditions). Thus, we presume that it this question Reviewer#3 means the ultimate physical limit in “ideal” noiseless conditions and absence of any other distortions in the initially acquired 3D set of OCT data. In such an ideal case, in principle, the well-known diffraction limit (or Abbe limit) of the focal-spot radius of lambda/2 is known. This ultimate result can be obtained for an illuminating beam with a very high numerical aperture (NA=>1), where NA is the numerical aperture that in reality is always smaller than unity. In practice, even in the absence of noises/distortions, the physical aperture D of the lens is always limited, which certainly limits the attainable effective NA and the degree of super-refocusing to a smaller value. There is a well know Abbe relationship for the attainable radius of the focal spot (i.e., the resolution limit for NA<1):  r=lambda/(2*NA). Certainly, optimization of parameters (including the regularization parameter alpha) improves the results of super-refocusing. However, as we pointed out in the text, it looks that in practice the initial resolution might be improved probably 2-3 times (depending on the initial value and the noise level), but still below the Abbe limit even for NA<1, first of all because of strong reduction in SNR. For improving the latter, one would be obliged to increase the regularization parameter, but this increase would spoil the resolution. Qualitatively, such statements are highlighted in the end of Section 5.2. But we think that it makes no sense to give specific numbers in addition to the data already shown in Figs. 6 and 7.

To conclude, we thank referee#3 and the other reviewers for the useful comments/suggestions. We did our best to incorporate them in the manuscript. These modifications were mentioned in our replies and are highlighted in the revised manuscript. We hope that we sufficiently clarified the questions of all referees, the revised manuscript is significantly improved and now it can be recommended for acceptance.

Round 2

Reviewer 3 Report (New Reviewer)

Comments and Suggestions for Authors

The authors have well-responded to my concerns in this revised manuscript. I think it can be considered for publication if the other reviewers have no objections.

This manuscript is a resubmission of an earlier submission. The following is a list of the peer review reports and author responses from that submission.

Round 1

Reviewer 1 Report

Comments and Suggestions for Authors

The article is very difficult to comprehend due to extensive grammatical errors and poor English quality. Furthermore, the article has to be reformatted to improve the quality of the presentation. Thus I suggest the authors to format the article accordingly and resubmit it. 

Comments on the Quality of English Language

The article is very difficult to comprehend due to poor English quality. Furthermore, the article has to be reformatted to improve the quality of the presentation. Thus I recommend rejection of this article. 

Reviewer 2 Report

Comments and Suggestions for Authors

The manuscript by A. L. Matveyev et al. derives equations to computationally transform data obtained in one OCT point-scanning geometry/setup to another point-scanning geometry (e.g., refocusing, Gaussian to Bessel beams, changing the numerical aperture, etc.). To do this, the authors derive expressions that require only one filter function, i.e., multiplication in 3D Fourier space. Overall, the manuscript is interesting and the equations and methods are properly derived for the most part. However, there are some shortcomings that should be corrected:

  • The "super-refocussing" part is not entirely new, and papers using very similar concepts should be cited and properly credited, for example: https://doi.org/10.1038/s41598-021-99889-3

  • The main message and achievements of this paper are a bit buried. For example, reading the abstract or even the introduction, I had no idea what to expect. Maybe the authors can give the examples discussed later (Gaussian to Bessel, computationally increasing NA, aberration correction and refocusing) in the abstract to illustrate the obtained filter function.

  • I am not sure of the motivation behind this approach, and the discussion is very limited. Previous works, such as ISAM and digital refocusing, were aimed at obtaining the unobscured view of the sample backscattering at diffraction-limited resolution, i.e. the ideally focused image data. The goal of this approach seems to be to translate to other limited imaging scenarios. For example, why would I want to go from a Gaussian beam to a Bessel beam instead of refocusing all depths? It will not have better resolution, but it will have all the side lobes associated with Bessel beams.

  • The aberration example states that the double-pass phase correction is insufficient. The authors should show the correct filter function. Does it include an amplitude component (which would affect the methods used to determine aberrations)? How far does it deviate from the double-pass phase function? Essentially: How much does it affect the previous methods?

  • The approach is only validated on simulated single point scatterers (i.e., essentially the depth-dependent PSF). It would be nice to see this demonstrated and validated on real image data. However, I realize that this may be beyond the scope of this paper.

  • Confocal gating is not really discussed or even mentioned. However, the change in actual sensitivity both in and out of focus when physically going from Gaussian to Bessel beams compared to their approach may be an important factor.

  • I am not quite sure why the scattered amplitude Ks is introduced. If it describes the backscattering of the field, it could technically depend on each scatterer. The authors should consider rephrasing that part and improve its clarity.

Minor remarks:

  • It is unclear to me why the manuscript starts with Monte Carlo simulations and drops the topic completely after the first paragraph.

  • Line 174ff: The authors mention that Us(x0, y0; k) explicitly depends not only on x0, y0 but also on xs, ys and even zs. Why is not given as Us(x0, y0, xs, ys, zs; k) then? Furthermore, the equation does not show a dependence on zs and being the illumination part it should not depend on zs from my understanding. For brevity, the authors could also summarize x0, y0 to a vector, etc. 

  • Line 186: The received spectral components are not explained, neither is that they follow a Gaussian distribution due to the confocality of the used single mode fiber (see above).

  • Line 220: Digital Fourier Transform is typically more commonly referred to as Discrete Fourier Transform. 

  • It is also a bit confusing that everything is given in terms of a continuous, analytical function, but when a Fourier transform is involved, it is the DFT rather than the Fourier transform.

  • Line 266: FFT should better be DFT for consistency. 

  • Line 486: non-Gausansi -> non-Gaussian

  • There is a huge amount of literature on holographic approaches and even use of k-space in OCT and scattering theory. The authors consistently cite their own research and the most important results from other groups, but it would be valuable to give a more complete picture.

Comments on the Quality of English Language
  • Overall, the manuscript can be understood. However, there are several typos, grammatical errors, and in particular a lot of nested sentences that make the manuscript difficult to read in places. This could be improved.

Reviewer 3 Report

Comments and Suggestions for Authors

The manuscript under review is dedicated to a very important problem of improving lateral spatial resolution of OCT systems by means of digital refocusing. The authors have performed an extensive theoretical analysis of OCT image formation in case of a highly scattering object under study. The analysis is based on the spatial frequency formalism and is aimed at solving the problem of lateral spatial resolution decrease at depths far from the imaging beam focus. The described refocusing approach is in fact equivalent to a number of deconvolution techniques, commonly used to improve spatial resolution of images. Importantly, the authors demonstrate how the regularization parameter, added to the denominator of filtering function in eq. (12) must be chosen according to the SNR value of the OCT image.

In my opinion, the only thing missing in the manuscript is a brief description of how the particular processed data was generated in Section 5.